# Asphalt Incorporation with Ethylene Vinyl Acetate (EVA) Copolymer and Natural Rubber (NR) Thermoplastic Vulcanizates (TPVs): Effects of TPV Gel Content on Physical and Rheological Properties

**DOI:** 10.3390/polym13091397

**Published:** 2021-04-26

**Authors:** Nappaphan Kunanusont, Boonchai Sangpetngam, Anongnat Somwangthanaroj

**Affiliations:** 1Department of Chemical Engineering, Faculty of Engineering, Chulalongkorn University, Bangkok 10330, Thailand; mai2newwood@hotmail.com; 2Department of Civil Engineering, Faculty of Engineering, Chulalongkorn University, Bangkok 10330, Thailand; boonchai.sa@chula.ac.th; 3Bio-Circular-Green Economy Technology & Engineering Center, BCGeTEC, Faculty of Engineering, Chulalongkorn University, Bangkok 10330, Thailand

**Keywords:** polymer modified asphalt, thermoplastic vulcanizate, gel content, natural rubber, ethylene vinyl acetate copolymer, rheological properties, storage stability

## Abstract

Plastic waste has been incorporated with asphalt to improve the physical properties of asphalt and alleviate the increasing trend of plastic waste being introduced into the environment. However, plastic waste comes in different types such as thermoplastic or thermoset, which results in varied properties of polymer modified asphalt (PMA). In this work, four thermoplastic vulcanizates (TPVs) were prepared using different peroxide concentrations to produce four formulations of gel content (with varying extent of crosslinked part) in order to imitate the variation of plastic waste. All four TPVs were then mixed with asphalt at 5 wt% thus producing four formulations of PMA, which went through physical, rheological, and storage stability assessments. PMA with higher gel content possessed lower penetration and higher softening temperature, indicating physically harder appearance of PMA. Superpave parameters remained unchanged among different gel content PMA at temperatures of 64, 70, and 76 °C. PMA with any level of gel content had lower Brookfield viscosity than PMA without gel content at a temperature of 135 °C. Higher gel content resulted in shorter storage stability measured with greater different softening temperatures between top and bottom layers of PMA after 5 days of 163 °C storage. This study shows that asphalt with thermoset plastic waste is harder and easier to pave, thus making the non-recycling thermoset plastic waste more useful and friendly to the environment.

## 1. Introduction

Plastics have been used in many applications since 1900s [1]. The accumulation of plastic waste has become a global concern, especially single-use plastic in the form of non-biodegradable plastic [2,3,4,5,6,7]. Plastics are often released into the sea either intentionally or unintentionally, and accumulation of such waste is considered to be harmful or even toxic to marine lifeforms. Many solutions have been developed to solve this problem with the method of utilizing plastic waste being one such option, for example depolymerization [8], the recycling process [9,10], or the blending of plastic waste with asphalt to prepare polymer modified asphalt [11,12,13,14,15,16,17]. The addition of plastics can improve the asphalt properties such as penetration, rutting resistance or Superpave parameter, and thermal stability.

Transportation is currently more convenient than ever. With cars being the main mode of transportation causing an increase in traffic load, the importance of good pavement is imperative. Polymer modified asphalt (PMA) has been introduced to improve the properties of pavement materials. The polymers can be classified into thermoplastic, thermoset, elastomer or rubber, and thermoplastic elastomer [18]. Thermoplastic such as polyethylene (PE) [19,20], polypropylene (PP) [21], polyethylene terephthalate (PET) [22] are used for the modification of asphalt, improving the rutting resistance and thermal cracking. However, due to the semi-crystallinity of thermoplastic, the elasticity of the material is still lacking at low temperature, thus limiting its applications. To improve the elasticity, an elastomer can be added to asphalt. Styrene butadiene copolymer (SBS) and styrene butadiene rubber are common elastomers used to improve asphalt’s properties [23,24,25,26]. Moreover, to utilize the waste, crumb rubber is also used to improve physical properties of asphalt [11,25,27,28]. The crumb rubber, which is in the form of fully crosslinked rubber, is added into the asphalt as particles resulting in a lower degree of processability and the possibility of phase separation occurring during storage if kept for a long duration. Therefore, the combination of both thermoplastic and vulcanized elastomer should be used to improve the asphalt [29,30].

Thermoplastic vulcanizate (TPV) is a material that has been widely used owing to its excellent elastic properties from the rubber particles and can be recycled like thermoplastic by melting [31,32]. Our previous work [33] has explored the suitable method for preparing thermoplastic vulcanizate from natural rubber and ethylene vinyl acetate copolymer to be used as a package of solidified asphalt. The “Split-DV” method was recommended to prepare the material pair that can be crosslinked using peroxide crosslinking agent. The aforementioned method also improved the compatibility of the involved materials by increasing degree of crosslink between plastic and rubber. The degree of crosslink of polymer can be described by gel content, which is the amount of insoluble sample remaining in hot solvent according to ASTM D2765 [34].

Moreover, there is a wide variety of plastic waste possessing different gel content, which affects the properties of PMA. To the best of our knowledge, there have been few studies focusing on the gel content of plastic waste before blending with asphalt. Rasool et al. [24] studied the effect of extrusion temperature on degraded tire blended with SBS and asphalt. High temperature resulted in more degraded rubber; thus, gel content of rubber decreased.

By imitating the plastic waste that contains various degrees of gel content, the present study focused on the effects of TPV gel content on properties of PMA. The TPV from natural rubber and ethylene vinyl acetate (EVA) copolymer with different gel content was mixed with asphalt to prepare PMA when the content of TPV was fixed at 5 wt%. Appearances and physical properties, i.e., penetration value, softening temperature, and rheological properties of PMA, were evaluated. In addition, the Brookfield viscosity was measured.

## 2. Materials and Methods

### 2.1. Materials

Ethylene-vinyl acetate (EVA) copolymer, which was used as the thermoplastic, was purchased from TPI Polene Co., Ltd. (Bangkok, Thailand). Natural rubber (NR) sheet (air-dried sheet type) was purchased from Bothong cooperation (Chonburi, Thailand). Dicumyl peroxide (DCP), which was used as a crosslinking agent, was supplied by Arkema (Missouri, North Kansas City, MO, USA). Octadecyl-3-(3,5-di-tert butyl-4-hydroxyphenyl)-propionate (Irganox1076) and tris(2,4-di-tert-butylphenyl)phosphate (Irgafos168), which were used as a thermal stabilizers, were purchased from Tokyo Chemical Industry Co., Ltd. (Tokyo, Japan). Asphalt binder (penetration grade, AC60/70) was obtained from Thai Lube Base Co., Ltd. (Bangkok, Thailand) Properties of the EVA, NR, DCP, and asphalt binder are shown in Table 1. Xylene, which was used as solvent to evaluate the gel content of thermoplastic vulcanizate, was used as received without further purification.

### 2.2. Sample Preparation

The sample preparation contained two steps: preparation of thermoplastic vulcanizates and preparation of polymer modified asphalt. The overall experimental procedure is shown in Figure 1.

#### 2.2.1. Preparation of Thermoplastic Vulcanizates (TPVs)

EVA pellets and NR sheets were dried in a vacuum oven at 60 °C for 4 h to remove moisture. The EVA/NR TPVs were prepared by melt-mixing in an internal mixer (Charoen Tut, Thailand) with chamber size of 60 cm^3^. According to our previous work [30], the suitable method used to prepare EVA/NR thermoplastic vulcanizate is the Split-DV method, which is shown in Figure 1a. Firstly, 25 wt% of EVA and 50 wt% of NR were pre-melted for 1 min and then dynamically vulcanized with the DCP for 5 min at 190 °C with a rotor speed of 60 rpm. After that, the product obtained from the previous step was melt-mixed with 25 wt% of EVA, and each thermal stabilizer (TS) of 0.5 phr was added at 130 °C with a rotor speed of 60 rpm for 6 min. DCP was varied from 0 to 1.5 phr to obtain the sample with different crosslinked degree. The samples were denoted as D0, D0.5, D1, and D1.5, and their appearance is shown in Figure 2. Gel content and swelling ratio of TPV was evaluated using the solvent extraction method according to ASTM D2765 [34]. The formulation, gel content, and swelling ratio of TPV are shown in Table 2.

#### 2.2.2. Preparation of Polymer Modified Asphalt (PMA)

The asphalt was melted at 150 °C in a metal container. Each TPV sample obtained from a previous section was cut into small pieces (≤2 mm) and gradually mixed with asphalt at 5 wt% using a high-speed shear mixer (L5M model, Silverson, Massachusetts, USA) at 150 °C with a rotor speed of 6000 rpm for 2 h and 3000 rpm for 2 h in order to remove air bubbles from asphalt. The protocol was adapted from Fang et al. [35]. The suitable condition to mix EVA and crumb rubber (CR) with asphalt were a mixing temperature of 140 °C, mixing time of 1.5 h, and rotor speed of 3750 rpm. However, our samples were 2 mm pieces of TPV. At first, in the use of this protocol, the TPV was not well-dispersed in the asphalt. Therefore, the protocol had to be adjusted by increasing the temperature, the mixing time, and the rotor speed as described above.

After that, the appearance of neat asphalt and PMA was evaluated by pouring the neat asphalt or obtained samples into water at room temperature. After the sample was solidified, the solid asphalt was then stretched by hand, and the texture was observed. Each PMA is represented as PMADx where Dx is the TPV sample blended with asphalt at 5 wt%. For example, PMAD0 is the asphalt modified with 5 wt% of D0 sample.

### 2.3. Characterizations

#### 2.3.1. Morphology

The morphology of PMA related to dispersion of TPV in asphalt binder was observed with a fluorescence microscope (DM 2500P, Leica) with a magnification of 20×. The sample was collected from the mixer and placed on a microscopic glass slide then covered with cover glass. The glass slide was placed in hot stage apparatus (T95-HS, Linkam, UK) and heated from room temperature to 150 °C and then held at constant temperature of 150 °C until the molten asphalt became a thin film under the cover glass. Photographs were taken using a digital camera attached to the microscope.

#### 2.3.2. Physical Properties

Physical properties such as penetration value of asphalt indicate the hardness of asphalt. The test was performed at 25 °C according to ASTM D5 [36]. Ring-and-ball softening temperature was measured according to ASTM D36 [37] with a heating rate of 5 °C/min.

#### 2.3.3. Performance Grading

Performance grade of neat asphalt and PMA was evaluated according to ASTM D7175 [38] using a dynamic shear rheometer (Physica MCR 501, Anton Paar, Graz, Austria) using a constant strain of 12% and frequency of 10 rad/s. The complex modulus (G*) and phase angle (δ) were measured at specific strain and angular frequency. Afterwards, the Superpave parameter (|G*|/sin δ) was calculated. The initial temperature of testing was 58 °C for neat asphalt and 64 °C for the PMA. The temperature was increased at an increment of 6 °C for each test until the value of the Superpave parameter (|G*|/sin δ) was lower than 1 kPa. Fail temperature of neat asphalt and PMA were calculated according to ASTM D7643 [39]. This Superpave parameter was introduced by the American Strategic Highway Research Program (SHRP). It is related to the stiffness of asphalt at a specific temperature. The Superpave parameter represents the property of asphalt in real traffic loading, which is usually in cycles.

#### 2.3.4. Rheological Properties

Rheological properties of neat asphalt and PMA such as storage modulus (G’), loss modulus (G”), phase angle (δ), and complex viscosity (η*) were measured using a dynamic shear rheometer (DSR) (Physica MCR 501, Anton Paar, Graz, Austria). The rheometer is a strain-controlled parallel-plate type. The test was performed in frequency-sweep mode with a plate diameter of 25 mm, with a gap of 1 mm at constant temperature of 60 °C (which is the road average surface temperature in hot climate region [23,40,41,42]) with angular frequencies of 100—0.1 rad/s and a constant strain of 12%, which is within the linear viscoelastic region of the asphalt (Appendix A).

#### 2.3.5. Brookfield Viscosity

Brookfield viscosities of neat asphalt and PMA were measured using a Brookfield viscometer (DVIII, Brookfield) at a constant shear rate of 18.6 s^−1^ (rotor speed of 20 rpm) using spindle 21 geometry. The temperature was set at 135 °C. The viscosities were recorded every 1 min from 0 to 10 min and every 5 min until 60 min. The equipment was set up according to ASTM D4402 [43].

#### 2.3.6. Storage Stability

Storage stability or separation tendency of polymer from PMA was evaluated according to ASTM D7173 [44]. This test was performed only on the asphalt blended with polymer. The asphalt sample was poured into an aluminum tube with a diameter of 25.4 mm and height of 136.7 mm. The tube was closed and stored in the oven at 163 °C for 5 days. The tube was cooled to −10 °C for 2 h and cut into three sections as shown in Appendix A. The difference of softening temperature of PMA collected from top and bottom sections of the tube was used to evaluate the storage stability of PMA.

## 3. Results

### 3.1. Appearance, Morphology, and Physical Properties

The appearance of neat asphalt and PMA at room temperature as well as their morphology at 150 °C observed with the fluorescence microscope are shown in Figure 3. It was found that the neat asphalt (Figure 3a) exhibited a glossy texture, while the PMAD0 exhibited a matte texture (Figure 3c). Only neat asphalt and PMAD0 exhibited smooth texture, while the PMAD0.5, PMAD1, and PMAD1.5 showed rough texture as can be seen in Figure 3e,g,i, respectively. The TPV consisted of thermoplastic and crosslinked rubber particles, which could be observed by SEM (Appendix A) from our previous work [33]. Therefore, the dispersion of TPV in PMA was observed in the fluorescence micrographs at 20× magnification.

The fluorescence micrograph of PMA exhibited heterogeneous morphology with different shape of dispersed phase (Figure 3d,f,h,j, respectively). The dispersed phase of PMAD0 was in a spherical shape, while those in PMAD0.5, PMAD1, and PMAD1.5 were in an irregular shape. Morphology of asphalt blended with TPV was similar to that blended with the crumb rubber because the crumb rubber also consisted of the crosslinked rubber [24,25,27]. Crosslinked rubber is like a thermoset plastic that cannot be melted at high temperature. Therefore, it only dispersed in the asphalt as observed.

Physical properties such as penetration and ring-and-ball softening temperature (TRB) are shown in Table 3. It was found that PMA exhibited less penetration of needles and higher softening temperature than neat asphalt. With the increase of gel content of TPV, the penetration decreased, and the softening temperature slightly increased. Moreover, higher gel content led to a higher degree of crosslinking in TPV causing the samples to be more rigid and difficult to deform, thus improving the asphalt properties. The improvement in penetration and softening temperature of modified asphalt indicated that the samples here are suitable for use in hot climate regions. However, these values represent only the properties in a static situation and do not reflect real world usage because roads normally experience continuous loading and unloading of forces during traffic. Therefore, the rheological properties of asphalt were characterized and are discussed in the next section.

### 3.2. Performance Grading

In this work, the neat asphalt and PMA were classified as Superpave performance grade classification. The Superpave parameters (|G*|/sin δ) at each temperature test and fail temperature are shown in Table 3. It was found that incorporation of TPV increased the fail temperature from 66.0 (neat asphalt) to 73.7 °C (PMAD1.5). Moreover, |G*|/sin δ of PMA was higher than that of neat asphalt by about 2.3 to 2.5 times. It can be said that the grade of modified asphalt was improved from the neat asphalt, and there was no significant difference between each PMA sample. Therefore, gel content of TPV has a small effect on the performance of asphalt compared to the content of TPV. However, the Superpave parameter was measured using constant frequency and amplitude. The real condition of traffic load must have both heavy and light load. Asphalt binder is a viscoelastic material. Therefore, the frequency sweep mode was used.

### 3.3. Rheological Properties

Rheological properties of neat asphalt and PMA in frequency sweep mode were evaluated to reflect the variation of force that the binder received during traffic transportation. Figure 4 shows rheological properties of neat asphalt and PMA and their values at frequencies of 0.1, 1, 10, and 100 rad/s, which are also summarized in Table 4. It was found that storage modulus (G’), loss modulus (G”), Superpave parameter (|G*|/sin δ), and complex viscosity (η*) of PMA were higher than those of neat asphalt, whereas phase angle (δ) of PMA was lower than that of neat asphalt in the test range, which indicated the improvement of asphalt by incorporation of TPV.

It was found that the G’ of PMAD0.5 was higher than that of other PMA at low frequency range (0.1–1 rad/s), while it was positively correlated with gel content at high frequency range (10–100 rad/s). The G” of PMA increased with increase of gel content for all tested frequency values. Moreover, the G” of all samples was higher than G’, and there was no crossover point between G’ and G”, which means that the behavior of PMA was viscous-dominant in the tested conditions. It can be observed that the loss modulus (Figure 4b), complex modulus (Appendix A), and |G*|/sin δ were similar. The phase angle (Figure 4c) of PMA was lower than that of neat asphalt. It was found that the PMAD0.5 showed the lowest phase angle at low angular frequency, while there was no significant difference at high angular frequency.

Complex viscosities of neat asphalt and PMA are shown in Figure 4d. The complex viscosity of neat asphalt was independent of angular frequency, indicating that the neat asphalt behaves as a Newtonian fluid in the testing range. However, the complex viscosity of PMA, which contained TPV, showed a frequency dependent behavior; the complex viscosity decreased with an increase in angular frequency. This behavior is non-Newtonian in nature with the type being pseudoplastic or shear thinning. The complex viscosity of PMA increased with increasing gel content of TPV at both low and high angular frequencies.

### 3.4. Brookfield Viscosity

Brookfield viscosity was measured using the Brookfield rotational viscometer at 135 °C. The viscosity of neat asphalt and PMA was recorded for 60 min as shown in Figure 5. The initial Brookfield viscosity value from the instrument was recorded at 2.500 Pa·s (2500 cP), which is the maximum value that can be measured by the instrument. The viscosities of neat asphalt and PMA decreased with time and remained stable after 10 min as shown in Figure 5. It was found that the Brookfield viscosity of PMA was higher than that of the neat asphalt due to the TPV restricting the mobility of asphalt. The PMAD1.5, which has the highest gel content, showed the lowest viscosity, while the PMAD0, which has no gel content, exhibited the highest viscosity among all the PMA samples.

### 3.5. Storage Stability

The storage stability of polymer modified asphalt was evaluated by the difference of softening temperature of asphalt collected from the top and bottom sections of the tested aluminum tube. The higher difference in temperature refers to more polymer separation from the asphalt. The density of polymer is lower than that of asphalt. Therefore, the polymer tends to float over the asphalt surface during separation. The values of softening temperature of PMA before and after storage at 163 °C for 5 days are shown in Table 5.

It was found that PMAD0 showed the lowest difference in softening temperature, while the PMAD1.5 showed the highest value.

## 4. Discussion

The content of TPV in PMA of the present study was fixed at 5 wt% (EVA 2.5 wt% and NR 2.5 wt%) because the asphalt is in continuous phase at this content, as can be observed from Figure 3. It would be convenient to observe the different shape of dispersed phase from TPV with different gel content. According to Sengoz et al. [45]’s work, the polymer entered continuous phase when content of polymer was more than 5 wt%.

Appearances of neat asphalt and PMAD0, which is the asphalt blended with D0, exhibited smooth surface while other PMAs exhibited rough surface. The physical properties of PMA such as penetration and softening temperature were affected by gel content of TPV. High gel content of TPV in PMA resulted in harder PMA as seen in the lower penetration value (higher degree of hardness) and higher softening temperature. This is consistent with the findings of Yan et al. [11] that crumb rubber (CR) content affected penetration and softening temperature more than that of EVA content. Changes of physical properties could probably be due to a different degree of crosslinking. According to Qian and Fan [25], addition of styrene–butadiene–styrene block copolymer (SBS) in rubberized asphalt containing 20 wt% CR can decrease the penetration value from 60 (0 wt% SBS) to 40 (3 wt% SBS). For EVA and CR modified asphalt, a penetration value of 40 can be observed from the asphalt modified with EVA 4 wt% and CR 10 wt% [11]. It seemed that TPV could achieve similar penetration from addition of only 5 wt% into asphalt.

The incorporation of TPV also increased the Superpave parameter (|G*|/sin δ) of PMA compared to the neat asphalt. However, the gel content of TPV had a small improvement effect on PMA in test conditions in which the Superpave parameter was measured using constant frequency and amplitude. The real condition of traffic load must have both heavy and light load. Asphalt binder is a viscoelastic material. Therefore, the rheological properties of neat asphalt and PMA in frequency sweep mode were analyzed.

Rheological properties of PMA at 60 °C such as storage modulus, loss modulus, Superpave parameter, and complex viscosity at high frequency range increased with the increase of gel content of TPV blended with asphalt, indicating that the material was more resistant to deformation due to crosslink in TPV. In this condition, the TPVs acted as particles to restrict the flow of PMA. For the low frequency range, the PMAD0.5 has the highest storage modulus and the lowest phase angle. According to the highest value of swelling ratio of D0.5 compared with other TPV, the asphalt might be absorbed in the 3D network of D0.5 more than D1 and D1.5. Therefore, the more swollen TPV showed more elastic behavior of polymer modified asphalt than the less swollen TPV. This result corresponded with the results from Yu et al. [46] that gel type bitumen possessed higher complex modulus and lower phase angle than the sol type bitumen. Although the frequency sweep test could provide preliminary information about PMA, to deeply investigate the fatigue resistance and elasticity performance of PMA, the linear amplitude sweep test (LAS) and multiple stress creep recovery (MSCR) should be further characterized.

As observed from Figure 4 (complex viscosity) and Figure 5 (Brookfield viscosity), the trend of viscosities of each sample measured by dynamic shear rheometer (DSR) was different from those measured by the Brookfield viscometer, which could be due to the different state of the polymer contained in the asphalt. The Brookfield viscosity was measured at a temperature higher than 135 °C, while the complex viscosity from DSR was measured at 60 °C. According to the melting temperature of EVA of 86 °C, the state of EVA in asphalt during the DSR test is solid, while that of EVA during the Brookfield test is molten. In the molten state, the polymer chains were dispersed in asphalt as a random coil. The TPVs with no crosslinked part (D0) were in the form of a long-chain polymer, which could entangle together, whereas the TPV with dynamically vulcanized fine rubber (D0.5, D1, D1.5) might have lower content of long-chain polymer compared with the D0 sample.

Figure 6a shows a schematic drawing of TPV dispersed in asphalt in the molten state, which had different gel content. The difference between long-chain and short-chain polymers was the degree of entanglement, which affected the flowability of the asphalt, with the short chain polymer resulting in low viscosity due to the low to non-existent degree of entanglement. The dispersion of TPV during the Brookfield viscosity measurement is illustrated in Figure 6b. Figure 6c shows the changes of microstructure of PMA when the shear force was applied. The long polymer chain could be stretched, and the entangled points restricted the movement, which needed shear stress to disentangle the polymer chains. According to previous studies [47,48], the media with high aspect ratio fiber was more viscous than that with low aspect ratio fiber as well as spherical particles. This is a hypothesis from an indirect result and some studies. The microstructure of PMA containing different structure of polymer at high temperature should be further investigated to clarify the explanation.

Storage stability of PMA was related to gel content and swelling ratio of TPV. As described in Table 2, the gel content of D1 and D1.5 is similar while the swelling ratio of D1.5 was lower than that of D1, showing that the D1.5 has a denser 3D network than the D1 sample. The dense 3D network could absorb less solvent. Therefore, the interaction between asphalt and D1.5 could be lower than other TPVs that had looser gel. This was consistent with Rasool and coworkers’ research showing that more crosslinked rubber showed lower compatibility with asphalt [24]. According to asphalt binder specification of the Department of Highways (Thailand) (Appendix A), the storage stability of natural rubber modified asphalt (NRMA) and polymer modified asphalt (PMA) are specified to be less than 4 and 2 °C when stored at 165 °C for 1 and 5 days, respectively. Therefore, the PMAD1.5 did not pass the specification of both PMA and NRMA and should not be used to prepare the PMA. It also did not pass the criteria of asphalt binder according to the specifications of the Department of Highways (Thailand).

## 5. Conclusions

Four thermoplastic vulcanizates of ethylene vinyl acetate copolymer and natural rubber with different gel content were assessed for physical and rheological properties and storage stability of polymer modified asphalt (PMA). Four formulations of PMA denoted as PMAD0, PMAD0.5, PMAD1, and PMAD1.5 were prepared with peroxide of 0, 0.5, 1, and 1.5 phr, respectively.

PMA with greater cross-linked TPV had a higher degree of hardness (penetration values of neat asphalt: 64, PMAD0: 47, D0.5: 46, D1: 45, and D1.5: 41) and higher softening temperatures (neat asphalt: 47.0 °C, PMAD0: 53.9 °C, D0.5: 54.5 °C, D1: 55.8 °C, and D1.5: 54.8 °C). The Superpave parameter (|G*|/sin δ) of PMA with any of TPV was higher than that of neat asphalt (neat asphalt: 1.3 kPa, PMAD0: 3.0 kPa, D0.5: 3.1 kPa, D1: 3.2 kPa, and D1.5: 3.3 kPa). This reflects the resistance to permanent deformation of PMA with TPV. Complex viscosities of PMA at a frequency of 10 rad/s were positively correlated with level of gel content (neat asphalt: 265 Pa·s, PMAD0: 615 Pa·s, D0.5: 641 Pa·s, D1: 770 Pa·s, and D1.5: 792 Pa·s), while Brookfield viscosity appeared differently. As gel content was increased, Brookfield viscosity decreased (neat asphalt: 0.300 Pa·s, PMAD0: 1.027 Pa·s, D0.5: 0.859 Pa·s, D1: 0.805 Pa·s, and D1.5: 0.802 Pa·s). As Brookfield viscosity was assessed at 135 °C, which is higher than the melting temperature of EVA at 86 °C, while complex viscosity was assessed at 60 °C, it is highly probable that melted EVA contributes to lower viscosity at high temperature. Adding more thermoset plastic to PMA could result in lower viscosity. Although Brookfield viscosity of neat asphalt is the lowest, neat asphalt has a lower Superpave parameter and degree of hardness. Adding thermoset plastic seems to improve physical and rheological properties of asphalt. The only disadvantage is worse storage stability of PMA, which challenges further investigations. We believe that adding thermoset plastic to form PMA has benefits of both improving asphalt and keeping the environment safer.

## Figures and Tables

**Figure 1 polymers-13-01397-f001:**
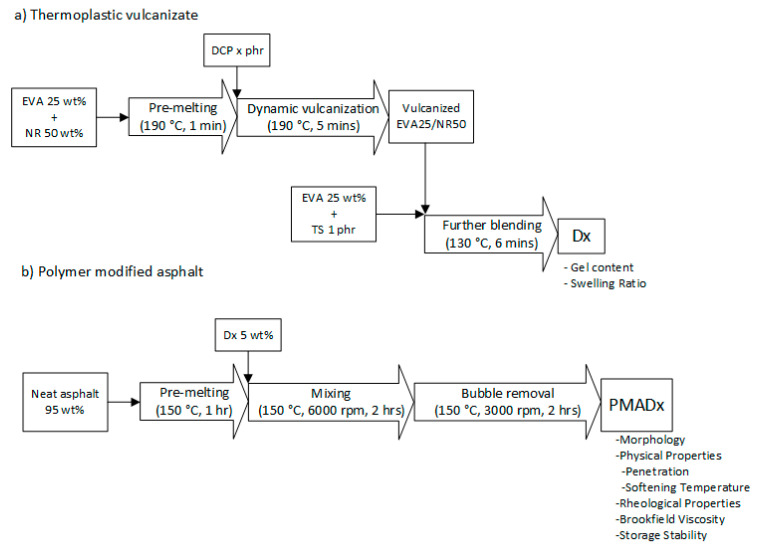
Experimental procedure of this research. (**a**) Preparation of thermoplastic vulcanizate; (**b**) preparation of polymer modified asphalt. “x” is DCP content.

**Figure 2 polymers-13-01397-f002:**
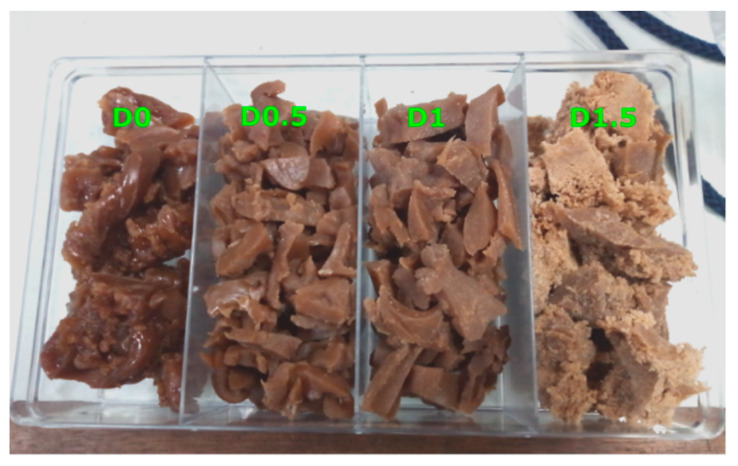
Appearance of EVA/NR thermoplastic vulcanizates at various DCP content.

**Figure 3 polymers-13-01397-f003:**
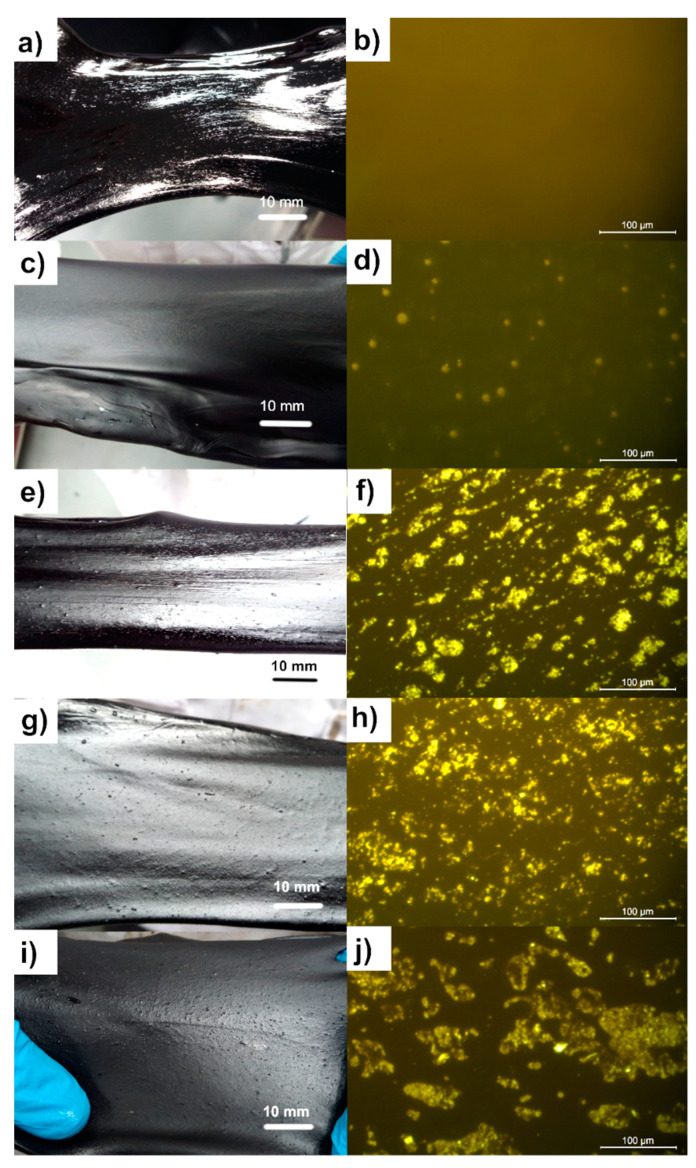
Optical images at room temperature and fluorescence micrographs at 150 °C of (**a**,**b**) neat asphalt, (**c**,**d**) PMAD0, (**e**,**f**) PMAD0.5, (**g**,**h**) PMAD1, and (**i**,**j**) PMAD1.5. The magnification of optical images and fluorescence micrographs were 1× and 20×, respectively.

**Figure 4 polymers-13-01397-f004:**
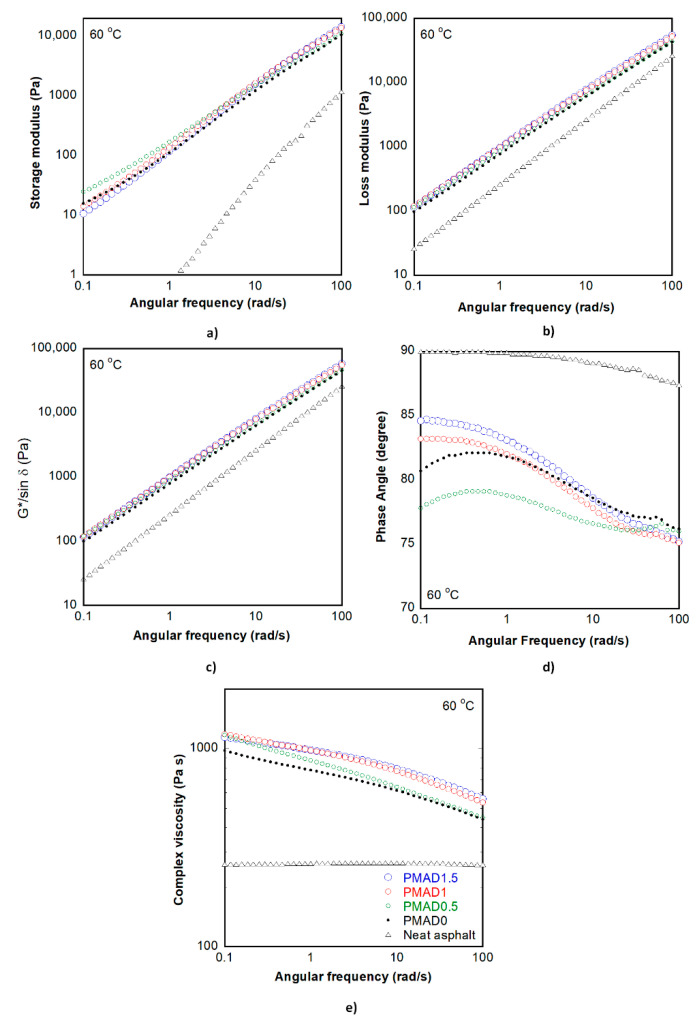
Rheological properties of neat asphalt and PMA at 60 °C in frequency sweep mode: (**a**) storage modulus, (**b**) loss modulus, (**c**) Superpave parameter, (**d**) phase angle, and (**e**) complex viscosity.

**Figure 5 polymers-13-01397-f005:**
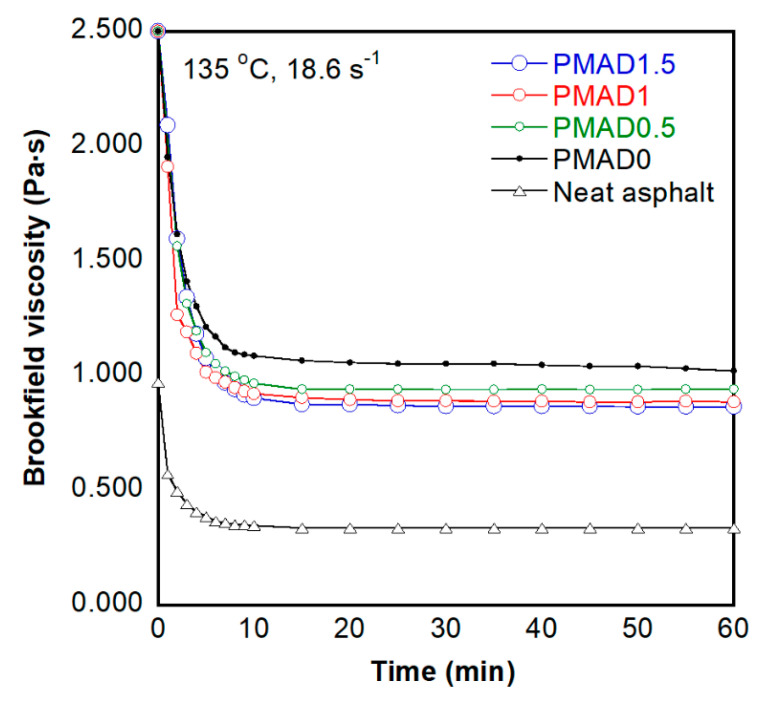
Brookfield viscosity of neat asphalt and PMA at 135 °C.

**Figure 6 polymers-13-01397-f006:**
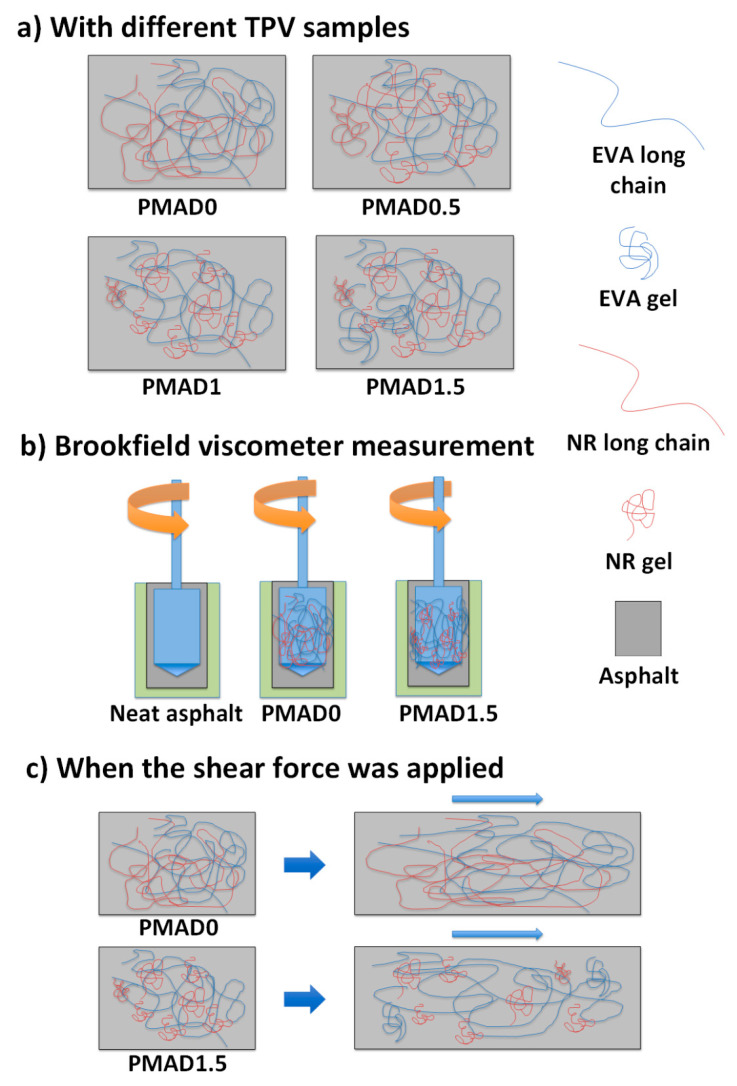
Schematic drawing of neat asphalt and PMA (**a**) with different TPV samples (**b**) during Brookfield viscosity measurement, and (**c**) when the shear force was applied to PMA samples.

**Table 1 polymers-13-01397-t001:** Properties of EVA, NR, DCP, and asphalt binder.

Properties	Unit	Value	Standard
**Ethylene Vinyl Acetate (EVA) Copolymer**			
Vinyl acetate content	wt%	18	ASTM D5594
Density	g/cm^3^	0.94	ASTM D1505
Melt flow index (2.16 kg, 190 °C)	g/10 min	2.3	ASTM D1238
Melting temperature	°C	86	ASTM D3418

**Natural Rubber (NR)**			
Mooney viscosity (100 °C)	ML(1 + 4)	59	ASTM D1646
Density	g/cm^3^	0.90	ASTM D792
Glass transition temperature	°C	−67	ASTM E1356

**Dicumyl Peroxide (DCP)**			
Purity	%	99.0	ASTM E755
Active Oxygen	%	5.9	ASTM D2180
Half-life temperature	°C		
10 h		117	
1 h		137	
1 min		178	

**Asphalt Binder (Penetration Grade, AC 60/70)**			
Penetration at 25 °C (100 g, 5 s)	0.1 mm	64	ASTM D5
Ring-and-ball softening temperature	°C	47	ASTM D36
Brookfield viscosity at 135 °C Shear rate 18.6 s^−1^, spindle 21	cP	297.5	ASTM D4402
	Pa s	0.2975	
|G*|/sin δ (10 rad/s, 12%strain) at	kPa		ASTM D7175
58 °C		3.06	
64 °C		1.29	
70 °C		0.592	
Fail temperature (at |G*|/sin δ = 1 kPa)	°C	66.0	

**Table 2 polymers-13-01397-t002:** Formulations as well as gel content and swelling ratio of EVA/NR thermoplastic vulcanizates.

Sample Name	Composition				Gel Content (%)	Swelling Ratio (%)
	EVA (wt%)	NR (wt%)	DCP (* phr)	TS (* phr)		
D0	50	50	0	1	Totally soluble	
D0.5	50	50	0.5	1	23.9 ± 2.9	1491.0 ± 45.8
D1	50	50	1	1	41.7 ± 1.3	1133.5 ± 48.9
D1.5	50	50	1.5	1	42.6 ± 0.6	1064.5 ± 16.2

* phr = parts per hundred resin.

**Table 3 polymers-13-01397-t003:** Physical properties as well as Superpave parameter of neat asphalt and polymer modified asphalt.

Sample Name	Penetration (0.1 mm) at 25 °C	T_RB_ (°C)	$\left(\frac{\left|G^{*}\right|}{\sin\delta }\right)(\mathrm{kPa})$				Fail Temperature (°C)
			58 °C	64 °C	70 °C	76 °C	
Neat asphalt	64 ± 0	47.0 ± 0.3	3.1	1.3	0.6	-	66.0
PMAD0	47 ± 1	53.9 ± 0.3	-	3.0	1.5	0.7	73.2
PMAD0.5	46 ± 1	54.5 ± 0.1	-	3.1	1.5	0.7	73.4
PMAD1	45 ± 1	55.8 ± 0.8	-	3.2	1.6	0.7	73.6
PMAD1.5	41 ± 1	54.8 ± 0.4	-	3.3	1.6	0.8	73.7

**Table 4 polymers-13-01397-t004:** Rheological properties of neat asphalt and PMA at 60 °C.

Parameters	Sample	Angular Frequency (rad/s)			
		ω = 0.1	ω = 1	ω = 10	ω = 100
Storage modulus (Pa)	Neat asphalt	0.020	0.599	40	1180
	PMAD0	15.8	111	1220	10,500
	PMAD0.5	24.7	169	1490	10,900
	PMAD1	14.1	136	1630	13,700
	PMAD1.5	10.7	118	1560	14,300
Loss modulus (Pa)	Neat asphalt	26	263	2650	26,000
	PMAD0	96.6	772	6030	42,700
	PMAD0.5	114	856	6240	43,500
	PMAD1	118	972	7520	51,700
	PMAD1.5	114	976	7770	54,100
G*/sin δ (Pa)	Neat asphalt	26.1	263	2650	26,000
	PMAD0	99.2	788	6280	45,300
	PMAD0.5	120	890	6590	46,200
	PMAD1	119	991	7880	55,300
	PMAD1.5	115	991	8080	57,900
Phase angle (degree)	Neat asphalt	90.0	89.9	89.1	87.4
	PMAD0	80.7	81.8	78.6	76.2
	PMAD0.5	77.8	78.8	76.6	76.0
	PMAD1	83.2	82.0	77.8	75.1
	PMAD1.5	84.6	83.1	78.6	75.2
Complex viscosity (Pa·s)	Neat asphalt	261	263	265	260
	PMAD0	979	780	615	440
	PMAD0.5	1170	873	641	448
	PMAD1	1180	981	770	535
	PMAD1.5	1140	984	792	559

**Table 5 polymers-13-01397-t005:** Storage stability of PMA contained different TPV after storage at 163 °C for 5 days.

Formula	Softening Temperature (°C)			
	Original	After Storage at 163 °C for 5 Days		
		Top	Bottom	Difference
Neat asphalt	47.0	-	-	-
PMAD0	53.9	55.4	53.6	1.8
PMAD0.5	54.5	57.2	55.0	2.2
PMAD1	55.8	58.5	56.1	2.4
PMAD1.5	54.8	60.8	52.1	8.7

## Data Availability

The data presented in this study are available on request from the corresponding author.

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
