# Peer review of "Asphalt Incorporation with Ethylene Vinyl Acetate (EVA) Copolymer and Natural Rubber (NR) Thermoplastic Vulcanizates (TPVs): Effects of TPV Gel Content on Physical and Rheological Properties"

_polymers, 2021, doi:10.3390/polym13091397_

Round 1

Reviewer 1 Report

This manuscript investigates the possibility to use plastic waste in bitumen modification by preparing thermoplastic vulcanizates (TPV) out of EVA copolymer and natural rubber. The effect of the gel content on the morphological, physical and rheological properties of TPV-modified bitumen is investigated.

This study certainly tackles an important challenge of using waste material to improve the performance of asphalt pavement materials. However, the manuscript should be improved before it can be published in the Journal. My comments and suggestions are listed below.

General comments:

  1. The authors conclude that the investigated materials are suitable for the use in asphalt pavements in hot climate regions. In particular, they emphasize that the use of high gel content TPV is recommended because the obtained asphalt exhibits superior physical and rheological properties as compared to bitumen modified with low gel content TPV. However, when concluding this, the authors completely ignore the fact that high gel content TPV is not compatible with bitumen, i.e. it possesses poor storage stability. It therefore should not be concluded that this material is suitable for paving applications as such.
  2. The authors demonstrate that bitumen modified with high gel content TPV tends to macro-phase separate at elevated temperatures. It could therefore be expected that homogeneous sample cannot be obtained when preparing the sample by melt blending. How did the authors ensure that the results obtained for the PMAD1.5 sample are representative (this applies to all microstructural, physical and rheological characterization)?
  3. The authors use the G* / sin δ parameter to describe the rutting resistance of the investigated samples. However, it has been shown that this parameter is not suitable for modified bitumen (see e.g. Delgadillo, R., Nam, K., & Bahia, H. (2006). Why do we need to change G*/sinδ and how?. Road materials and pavement design7(1), 7-27.). I would recommend using the MSCR test to evaluate the rutting resistance of the investigated samples.
  4. Based on the measurements of physical and rheological properties, the gel content of TPV has only a minor effect on the performance of TPV modified bitumen. The authors should address this in the manuscript, stating that the gel content has a much smaller effect on the binder performance than the amount of modifier, for example. 
  5. The language of the manuscript should be improved. It contains several typos and sentences that are convoluted / difficult to understand. Here are a few examples of poor English:
    Lines 51-55: "Although crumb rubber additives can be used to improve the properties of asphalt, it is without its own shortcoming. Due to crumb rubber already coming in the form of fully crosslinked rubber, the additive existed as a dispersion in the asphalt blend resulting in the lower degree of processability and the possibility of phase separation occurring during storage if kept for a long duration."
    Lines 234-237: "During traffic, the road experienced continuous loading and unloading of forces which may have a significant impact on the rheological properties of asphalt, as such the characteristics of the materials under such situation were further investigated."
    Lines 270-272: "As mentioned in the previous section, more gel content of TPV attributed to the more degree of crosslink of rubber particles which was difficult to deform."
  6. The authors always refer to figures and tables in past tense (e.g. "was shown in Figure 1", "were shown in Table 2"). Please use present tense in these instances instead.
  7. The authors should spell out abbreviations only when they are used the first time. For example, the abbreviation "polymer modified asphalt (PMA)" has been defined ten times in the manuscript, which is completely unnecessary.
  8. Throughout the manuscript, the unit of the rutting parameter is missing: it should be "G*/sin δ = 1 kPa", not "G*/sin δ = 1".
  9. The authors use two viscosity units in the manuscript (cP and Pa.s). It would be clearer if the authors would use only one viscosity unit consistently.

Detailed comments:

Lines 58-59: "According to the authors’ knowledge, TPV has only been utilized in one research with promising result [23];" I believe that the authors intend to say that there is only one research where TPV has been used as bitumen modifier.

Line 113: Please explain what "phr" means.

Sections 2.3.2. and 2.3.3.: Penetration and Ring-and-Ball softening point tests are standardized test methods. It is sufficient to refer to the relevant standards (ASTM D5 and ASTM D113) for the description of these tests. Sections 2.3.2. and 2.3.3. can be removed as they do not provide any additional information.

Line 170: The authors should clarify how it was determined that 12 % strain was within the linear viscoelastic region of the investigated samples. Did the authors perform strain sweep measurements for all the samples?

Lines 198-208: This paragraph is unnecessary, it can be removed (all this information has been provided already earlier in the manuscript).

Lines 222-224: "The microstructure of asphalt blended with TPV was similar to that blended with the crumb rubber because the crumb rubber also consisted of the crosslinked rubber." Please provide a reference to a study where a similar microstructure has been observed in crumb rubber modified bitumen.

Lines 245-250: This paragraph is unnecessary, it can be removed (rheological test methods have been described already in the Materials and Methods section).

Lines 260-263: The authors speculate that the elasticity of the investigated PMA samples is related to the swelling of TPV. The authors should comment if these differences in the swelling are observable in the micrographs of Figure 4.

Lines 278-283: The measurement of the rheological properties and the rutting parameter G* / sin δ is described already for the third time here, this is unnecessary repetition.

Figure 6: The authors have used linear interpolation to determine the temperature at which G* / sin δ equals 1 kPa. However, according to the ASTM D7643 standard, a different type of interpolation should be used to determine the fail temperature. Please revise accordingly.

Line 333: It is not clear to me how the effect of the aspect ratio of fibers on their viscosity is related to the viscosity changes in the investigated TPV modified bitumen samples. 

Reviewer 2 Report

The authors reported the modification of asphalt using EVA/NR TPV.

On the positive side, the properties of asphalt, such as hardness, softening temperature and rheological properties, were improved upon modification. The improvements were found to be dependent upon the gel content of TPV.

On the negative side, it is not an in-depth study as there are a number of issues unanswered. Firstly, the TPV was based on the ratio of 50/50 EVA/NR. Would a change in the ratio (for example 25/75 or 75/25) bring along an even better improvement?

Secondly, the TPV content was fixed at 5wt%. Similarly, would an increase in TPV content bring along better improvement? In other words, what the authors found may not represent the best formulation.

Thirdly, no comparison was made between this work and other related works previously reported in the literature. For example, reference 9 dealt with the modification of asphalt using waste rubber tires and EVA. Reference 19 dealt with asphalt modified with crumb rubber and SBS. Is the PMA developed by the authors any better than those reported in References 9 and 19?

In view of the shortcomings of the present work, the reviewer does not recommend the acceptance of the manuscript.

Reviewer 3 Report

Evaluation:

- The title does not clearly and sufficiently reflect the content of the article.

- Line 13: "... improve the properties of Asphalt ..." - Which properties?

- Lines 18-19: “... used to improve the properties of 18 asphalt.” - What properties?

- Abstract: the research method needs to be briefly described.

- Keywords: the words are well indexed.

- Lines 32-78 - 1. Introduction: the introduction is not adequate. The contribution of paper to the area of knowledge is unclear. The authors also do not present the current state of knowledge on the subject to be investigated. Little information and results of studies already developed are presented. The purpose of the article is also not well defined in the introduction. It is essential that the introduction has: appropriate contextualization of the problem to be studied; description of the current state of knowledge on the subject; support for the purpose of the study; theoretical bases clearly specified.

- Lines 81-89: further characterization information is required in relation to the following materials: ethylene-vinyl acetate copolymer (EVA), natural rubber sheet (NR) and dicumyl peroxide (DCP).

- Line 89: why was this asphalt binder (penetration grade, AC60 / 70) chosen?

- Line 110: why was 25 wt% of EVA and 50 wt% of NR used? Justify.

- Tables 2 and 3: justify the proportions (formulations) adopted in the research.

- 2.2 Sample preparation (lines 96-137): the procedure is not clear. Especially regarding the parameters (temperature, time, rpm, etc.) adopted in the preparation. Why were these values adopted for the parameters? At some point, the authors cite a previous work, however, readers may not have access to this publication.

- 2.3 Characterizations (lines 139-195): the following tests were carried out on the formulations proposed in the study: fluorescence microscope; penetration; ring-and-ball softening; storage modulus, loss modulus, phase angle and complex viscosity; Brookfield viscosity; storage stability. It is possible to verify that important tests were not conducted, such as: LAS (Linear Amplitude Sweep) and MSCR (Multiple Stress Creep Recovery).

- Figure 3: this figure is unnecessary.

- 3.1 Appearance, microstructure, and physical properties of neat asphalt and PMA (lines 197-242): the authors mention that they are investigating the microstructure from optical images and fluorescence micrographs. However, only with this technique it is not possible to fully study the microstructure of the material in question.

- 3.2 Rheological properties (lines 244-295): quite simple results. The authors should have evolved in rheological analyzes.

- 3.3 Brookfield viscosity (lines 297-339): in this section, several conclusions presented by the authors, cannot be confirmed by the data of the conducted tests, they are conclusions without confirmation by scientific method. Check point to point in this section.

- 3.4 Storage stability (lines 340-357): what are the tolerable limits for this test? acceptance criteria? Make this discussion.

- Regarding the discussion of the results, in general, the authors discuss the results superficially and do not compare the results obtained with other studies already published.

- The conclusions presented by the paper are quite simple and known to researchers in the field.

Round 2

Reviewer 1 Report

I suggest the publication of the manuscript in the present form.

Reviewer 2 Report

The authors have made substantial changes in the manuscript following the reviewer's comments. The reviewer recommends the acceptance of the manuscript.

Reviewer 3 Report

The authors did a good job review. The article has improved substantially.

All the questions below, which were submitted in the first version of the article, were answered appropriately, so I recommend publishing the article.

Questions of the first version of the article:

- The title does not clearly and sufficiently reflect the content of the article.

- Line 13: "... improve the properties of Asphalt ..." - Which properties?

- Lines 18-19: “... used to improve the properties of 18 asphalt.” - What properties?

- Abstract: the research method needs to be briefly described.

- Keywords: the words are well indexed.

- Lines 32-78 - 1. Introduction: the introduction is not adequate. The contribution of paper to the area of knowledge is unclear. The authors also do not present the current state of knowledge on the subject to be investigated. Little information and results of studies already developed are presented. The purpose of the article is also not well defined in the introduction. It is essential that the introduction has: appropriate contextualization of the problem to be studied; description of the current state of knowledge on the subject; support for the purpose of the study; theoretical bases clearly specified.

- Lines 81-89: further characterization information is required in relation to the following materials: ethylene-vinyl acetate copolymer (EVA), natural rubber sheet (NR) and dicumyl peroxide (DCP).

- Line 89: why was this asphalt binder (penetration grade, AC60 / 70) chosen?

- Line 110: why was 25 wt% of EVA and 50 wt% of NR used? Justify.

- Tables 2 and 3: justify the proportions (formulations) adopted in the research.

- 2.2 Sample preparation (lines 96-137): the procedure is not clear. Especially regarding the parameters (temperature, time, rpm, etc.) adopted in the preparation. Why were these values adopted for the parameters? At some point, the authors cite a previous work, however, readers may not have access to this publication.

- 2.3 Characterizations (lines 139-195): the following tests were carried out on the formulations proposed in the study: fluorescence microscope; penetration; ring-and-ball softening; storage modulus, loss modulus, phase angle and complex viscosity; Brookfield viscosity; storage stability. It is possible to verify that important tests were not conducted, such as: LAS (Linear Amplitude Sweep) and MSCR (Multiple Stress Creep Recovery).

- Figure 3: this figure is unnecessary.

- 3.1 Appearance, microstructure, and physical properties of neat asphalt and PMA (lines 197-242): the authors mention that they are investigating the microstructure from optical images and fluorescence micrographs. However, only with this technique it is not possible to fully study the microstructure of the material in question.

- 3.2 Rheological properties (lines 244-295): quite simple results. The authors should have evolved in rheological analyzes.

- 3.3 Brookfield viscosity (lines 297-339): in this section, several conclusions presented by the authors, cannot be confirmed by the data of the conducted tests, they are conclusions without confirmation by scientific method. Check point to point in this section.

- 3.4 Storage stability (lines 340-357): what are the tolerable limits for this test? acceptance criteria? Make this discussion.

- Regarding the discussion of the results, in general, the authors discuss the results superficially and do not compare the results obtained with other studies already published.

- The conclusions presented by the paper are quite simple and known to researchers in the field.